# First Principles Investigation of C, Cl_2_ and CO Co-Adsorption on ZrSiO_4_ Surfaces for Carbochlorination Reaction

**DOI:** 10.3390/ma17071500

**Published:** 2024-03-26

**Authors:** Xingping Liu, Fumin Wang, Yalan Zhao, Arepati Azhati, Xingtao Wang, Zhengliang Zhang, Xueqian Lv

**Affiliations:** 1School of Chemical Engineering and Technology, Tianjin University, Tianjin 300350, China; tgwxt@tju.edu.cn (X.W.); zzl@tju.edu.cn (Z.Z.); 2Xinte Energy Co., Ltd., Urmqi 830011, China; zhaoyalan@xinteenergy.com (Y.Z.); arepati@xinteenergy.com (A.A.);

**Keywords:** first principles, co-adsorption, carbochlorination reaction, ZrSiO_4_

## Abstract

The study of the adsorption behavior of C, CO and Cl_2_ on the surface of ZrSiO_4_ is of great significance for the formulation of the technological parameters in the carbochlorination reaction process. Based on first principles, the adsorption structure, adsorption energy, Barder charge, differential charge density, partial density of states and energy barrier were calculated to research the adsorption and reaction mechanism of C and Cl_2_ on ZrSiO_4_ surfaces. The results indicated that when C, CO and Cl_2_ co-adsorbed on the surface of ZrSiO_4_, they interacted with surface atoms and the charge transfer occurred. The Cl_2_ molecules dissociated and formed Zr-Cl bonds, while C atoms formed C1=O1 bonds with O atoms. Compared with CO, the co-adsorption energy and reaction energy barrier of C and Cl_2_ are lower, and the higher the C content, the lower the adsorption energy and energy barrier, which is beneficial for promoting charge transfer and the dissociation of Cl_2_. The 110-2C-2Cl_2_ has the lowest adsorption energy and the highest reaction activity, with adsorption energy and energy barriers of −13.45 eV and 0.02 eV. The electrons released by C are 2.30 e, while the electrons accepted by Cl_2_ are 2.37 e.

## 1. Introduction

Zirconium oxychloride (ZrOCl_2_) is an important intermediate compound in the zirconium industry, and is an important raw material for preparing high-purity zirconium, zirconia powder, catalysts, etc., being widely used in ceramics, chemical industry, electronics, military and other fields [1,2,3,4,5,6,7,8,9]. Zirconium oxychloride is generally prepared by the alkali melting of ZrSiO_4_ [10,11] or hydrolysis of ZrCl_4_ [12]. Compared with the alkali melting method, the ZrCl_4_ hydrolysis method is more environmentally friendly, and has been an important method for preparing high-quality zirconium oxychloride (ZrOCl_2_).

Zircon sand (ZrSiO_4_), as a mineral resource with abundant reserves, which was one of the main raw materials for the preparation of zirconium tetrachloride, and the by-product (silicon tetrachloride) produced during the carbochlorination process of ZrSiO_4_, can be used as an important raw material for fumed silica and trichlorosilane. Therefore, compared with desiliconized zirconia and zirconium carbonitride, the carbochlorination of ZrSiO_4_ is a route with higher economic benefits and better development prospects.

With the continuous development of computer technology, first principles calculations have been widely applied in the exploration of reaction mechanisms [13,14,15,16] at the molecular level. In the field of carbochlorination, researchers revealed the reaction mechanism of the carbochlorination of TiO_2_ on the atomic adsorption level. For example, DUNE et al. [17] found in the reaction of C with Cl_2_-Ti that the Cl_2_ molecules adsorbed on the surface of C to generate the Cl•, which exists in the form of Cl atoms or Cl_X_ molecules. The active chlorine migrate towards the Ti-C interface and reacts to form titanium chlorides. Doris Vogtenhuber [18] and Oliver [19] investigated the adsorption reaction of Cl atoms on O defects and intact (110) surfaces of TiO_2_ using density functional theory (DFT), which indicated that compared with the intact TiO_2_ (110) surface, the adsorption activity of surface bridging oxygen vacancies and other defects is higher, making it easier to adsorb Cl atoms. Yang et al. [20] analyzed the adsorption reactions of C and Cl_2_ on the (100) plane of TiO_2_, and found that Cl atoms were unable to form bonds on the (100) plane, while C and Cl_2_ can form bonds on the (100) plane of TiO_2_. Compared with CO, the stability of the adsorption structure of C and Cl_2_ on the TiO_2_ (100) surface was higher, and C was more conducive to promoting the adsorption of Cl_2_ on the TiO_2_ surface. During the chlorination process of zirconium silicate, reducing agents need to be added to reduce the activity and reaction free energy of the product oxygen, and accelerate the reaction rate; however, due to the complexity of the reaction system, there have been no reports on the boiling chlorination reaction mechanism of ZrSiO_4_, especially the co-adsorption/reaction mechanism of the reducing agent C on Cl_2_ on the surface of zirconium silicate at the atomic scale.

The crystal structure of zirconium silicate (ZrSiO_4_) belongs to the tetragonal crystal system (ABO_4_ structures) [21]; compared with other crystal planes, (100) and (110) have lower crystal plane indices, larger interplanar spacing, more atoms arranged on this crystal plane, lower surface energy, and more adsorption active sites [22]. Thus, this article focused on the (100) and (110) surfaces as research objects, investigating the co-adsorption behavior of C, CO and Cl_2_ on the (100) and (110) surfaces of ZrSiO_4_ at the atomic level. Based on DFT calculations, the positive effect of C was clarified, which can provide assistance in understanding the reaction mechanism of ZrSiO_4_ carbochlorination.

## 2. Calculation and Experimental Methods

In this work, all calculations were conducted within the framework of density functional theory, using the projection enhanced plane wave method implemented in the VASP simulation software (VASP 6.3.2, Quantum Wise, Copenhagen, Denmark) [23]. The generalized gradient approximation method proposed by Perdew, Burke and Ernzerhof was chosen as the exchange correlation potential [24]. The long-range van der Waals interactions were described by DFT-D3 [25]. The cutoff energy of the plane wave was set to 480 eV. In the iterative solution of the Kohn Sham equation, the energy standard is set to 10^−5^ eV.

The 15 Å vacuum layer was added above the plane perpendicular to the lattice to avoid artificial interactions between periodic images. For the energy calculation of the (110) and (100) surfaces of ZrSiO_4_, the Brillouin zone was collected using the K points of 3 × 2 × 1 and 3 × 3 × 1. All structures were relaxed before the force on residual atoms dropped below 0.05 eV/Å. To calculate the density of states, the K points at the Brillouin zone were increased to 5 × 5 × 1 and 4 × 3 × 1, respectively. The transition states were searched by climbing image assisted elastic band (CI-NEB) [26,27]. During the transition state search, the minimum energy path of all images, including the initial and final states, were checked. The force convergence standard for each free atom was set to 0.05 eV Å^−1^.

The mixed powder of ZrSiO_4_ and petroleum coke was heat-treated under vacuum conditions, the heat treatment condition was 1473 K/1h, and the vacuum degree was higher than 10^−2^ Pa. The morphology and elemental composition of the powder were observed with field emission scanning electron microscopy (SU8010). Under a nitrogen atmosphere, the weight loss of the powder was measured using a thermogravimetric differential thermal analyzer (HITACHI STA7300, HITACHI, Tokyo, Japan), with a temperature range of 298–1473 K and a heating rate of 10 K/min.

## 3. Results and Discussion

### 3.1. Modeling of ZrSiO_4_ Structure

The crystal structure of zirconium silicate belongs to the tetragonal crystal system, with cell parameters of a = b = 6.631 Å, c = 5.983 Å, and it present with a structure which has eight-coordinated Zr and four-coordinated Si atoms. Zircon has the lowest density among the structures with Zr/Si = 8/4 coordination [21]. In this article, the K point on the ZrSiO_4_ (100) surface was set to 2 × 2 × 1 and the cutoff energy was set to 480 eV (Appendix A). The K point and the cutoff energy on the ZrSiO_4_ (110) surface were set to 3 × 2 × 1 and 480 eV, respectively (Appendix A). The optimized structure showed that the surface of ZrSiO_4_ is composed of Zr (green ball), Si (blue ball) and O (red ball) atoms, which were 4-coordinated Zr atoms, four-coordinated Si atoms, three-coordinated O atoms and two-coordinated O atoms, respectively (Appendix A).

### 3.2. Adsorption Behavior of C, CO and Cl_2_ on the (100) and (110) Planes

Herein, the co-adsorption behavior of C, CO and Cl_2_ on the (100) and (110) planes and their adsorption behaviors are described. According to the optimized adsorption model (Appendix A, gray balls represent C atoms and pink balls represent Cl atoms), differential charge density (Appendix A, blue and yellow regions represent charge accumulation and charge depletion, respectively) and Bader charge (Appendix A) results, it can be seen that both C and Cl_2_ molecules could form bonds with surface atoms when adsorbed on (100) and (110) planes individually, but CO could not. The C and Cl atoms had the ability to contribute electrons, while O and Zr atoms had the ability to accept electrons. The adsorption energy of C was higher than CO and Cl_2_. The adsorption energies of C, CO and Cl_2_ on the (100) plane were −3.01 eV, −0.48 eV and −0.39 eV, respectively, and the adsorption energy on the (110) plane changes to −3.69 eV, −0.16 eV and −0.53 eV. As a comparison, this article also differentiates between C, CO and Cl_2_ and the surface of zirconium silicate (before adsorption) was simulated by DFT, as shown in Appendix A.

### 3.3. Co-Adsorption Behaviors of C, CO and Cl_2_ on the (100) Plane

Figure 1 shows the co-adsorption structure of C and Cl_2_ (Figure 1a,c), as well as CO and Cl_2_ (Figure 1b,d), on ZrSiO_4_ (100). When the C, CO and Cl_2_ were co-adsorbed on the (100) plane, the chemical bonds between Cl_2_ molecules were extended to 3.59 Å and 3.55 Å, respectively, and then the activated molecules dissociated into two Cl atoms and formed chemical bonds with Zr atoms. The interatomic spacing of the chemical bonds formed with Zr atoms are 2.44 Å and 2.59 Å, respectively, while the interatomic spacing of the chemical bonds were extended to 2.46 Å and 3.01 Å when Cl_2_ and CO were co-adsorbed on the (100) plane. When C atoms were adsorbed on the (100) plane, the Zr1-O1 bond on the plane was broken, forming a C1=O1 bond with an interatomic spacing of 1.221 Å.

The interatomic spacing of the Si1-O1 bond was extended to 1.80 Å, and the distance between Zr1 and O1 was extended to 3.22 Å. After the CO molecules were adsorbed on the (100) plane, the Zr-O bonds were broken and the interatomic spacing increased to 2.39 Å. The chemical bond between the CO and O was formed with an interatomic spacing of 1.34 Å, accompanied by a similar phenomenon of O atoms protruding upward from the surface. These results show that when C, CO and Cl_2_ were co-adsorbed on the (100) plane, the binding strength of the Zr-O and Si-O bonds was weakened; accompanied by the formation of O-C and Zr-Cl bonds, the addition of C and CO can promote the dissociation of Cl_2_, reduce the strength of the surface Zr-O bonds and enhance the interaction between Cl_2_ and the (100) plane. Compared with CO (−2.65 eV), the co-adsorption energy of C and Cl_2_ on the (100) plane was lower (−6.61 eV), indicating that the co-adsorption structure formed by C atoms and Cl_2_ molecules on the (100) plane was more stable, and that elemental C was more conducive to promoting the dissociation of Cl_2_.

Table 1 shows the Bader charges of the C, CO and Cl_2_ adsorbed on the (100) plane, with C atoms as electron providers and Cl and O atoms as electron acceptors. In the 100-C-Cl_2_ structure, C recaptured the surface O to form CO molecules, and the number of electrons released by C towards the (100) surface was 1.06 e. When Cl atoms interacted with (100) plane, the electrons captured from the surface were 0.44 e and 0.62 e, respectively. For the 100-CO-Cl_2_ structure, the electrons accepted by the O atom from the (100) plane were 1.05 e, the electrons released by the C atom to the (100) plane were 1.78 e and the electrons captured by Cl atoms from the (100) plane were 0.16 e and 0.64 e, respectively, indicating that when CO and Cl_2_ were co-adsorbed on the (100) plane, CO molecules detached O from the (100) plane to form CO_2_ molecules, the electrons were released from the (100) plane and there was a strong interaction between CO molecules and the (100) plane.

Figure 2 shows the differential charge density diagram of C and Cl_2_ (Figure 2a,c), as well as CO and Cl_2_ (Figure 2b,d) co-adsorption structures on the (100) plane. When the C and Cl_2_ were co-adsorbed on the (100) plane, the Zr-Cl and C-O bonds were formed. A blue area representing charge accumulation was observed around the C-O-Si bond, indicating that C outputs electrons during the bonding process. The Cl_2_ molecule was dissociated into two Cl atoms, forming Zr-Cl bonds with surface Zr atoms, respectively. There is a clear yellow area representing charge depletion around the Zr-Cl bond, indicating a strong interaction between Zr atoms and Cl atoms. Similar results were observed when CO and Cl_2_ were co-adsorbed on the (100) plane; the above results indicated that the addition of C and CO could promote a charge transfer between the Cl_2_ and the (100) plane, and enhance the interaction between them.

Figure 3 shows the partial density of states of C and Cl_2_ (Figure 3), as well as that of CO and Cl_2_ (Figure 3b) co-adsorbed on the (100) plane, where the Fermi level energy was set to 0 eV. When C and Cl_2_ were co-adsorbed on the (100) plane, resonance peaks were observed at −1.35 eV, −0.97 eV, 3.81 eV, 5.91 eV and 6.10 eV, and were related to the bonding effect between Zr and Cl atoms. The resonance peaks related to C and O atoms were observed at −7.15 eV, −6.97 eV, −0.97 eV, 3.81 eV and 4.00 eV, while the resonance peaks which related to Si -O bonds were −7.92 eV, −6.00 eV, and −5.56, 5.91 eV and 8.27 eV, respectively. When CO and Cl_2_ molecules were adsorbed on the (100) plane, the resonance peaks related to Zr-Cl bonds shifted to −16.94 eV, −0.29 eV, 3.72 eV, 4.67 eV and 5.88 eV. The resonance peaks related to the bonding of C and O atoms were −7.92 eV, −6.96 eV, −2.38 eV, 3.46 eV and 3.72 eV, while the resonance peaks related to the bonding of Si and O atoms were −16.94 eV, −6.00 eV, −4.41 eV, 4.67 eV and 8.29 eV. The results indicated that Zr and Cl atoms and C and O atoms, as well as Si and O atoms, are prone to form chemical bonds, and the bonding interactions are strong.

Figure 4 shows the reaction pathways and energy barrier of C and Cl_2_ (Figure 4a–c) as well as CO and Cl_2_ (Figure 4d–f) co-adsorbed on the (100) plane. The adsorption energy of C and Cl_2_, as well as CO and Cl_2_, on the (100) surface to form the initial structure (IS) was set to 0 eV, which served as the energy for CO and CO_2_ to be desorbed from the initial state on the (100) plane, respectively. When C and Cl_2_ were adsorbed on the (100) plane, the C and O atoms overcame an energy barrier of 0.22 eV to reach a transition state, then CO was generated, accompanied by the release of 3.59 eV energy. When CO and Cl_2_ were adsorbed on the (100) plane, the process of reaching the transition state required overcoming 1.32 eV of energy, and the CO_2_ was formed. Subsequently, 3.59 eV of energy was released to reach the final state (FS), and CO_2_ was desorbed from the (100) plane. Compared with CO, the formation of CO_2_ was more difficult, with a higher energy barrier, and the C atom was more conducive to promoting the chlorination reaction of Cl_2_ molecules on the (100) plane.

Based on the above results, the co-adsorption structure formed by C atoms and Cl_2_ molecules on the (100) plane was more stable, and the energy barrier that needed to be overcome to reach the transition state was lower, compared with CO, which was conductive to promoting the charge transfer between the Cl_2_ and (100) plane, enhancing the interaction between them.

### 3.4. Co-Adsorption Behavior of C, CO and Cl_2_ on the (110) Plane

#### 3.4.1. Co-Adsorption on the (110) Plane with a Molar Ratio of C to Cl_2_ of 1:1

Figure 5 is the co-adsorption structure of C and Cl_2_ (a and c), as well as CO and Cl_2_ (b and d), on the (110) plane with a molar ratio of C to Cl_2_ of 1:1. The adsorption energies of C, CO and Cl_2_ on the (110) plane were −3.69 eV (Appendix A), −0.16 eV (Appendix A) and −0.53 eV (Appendix A), respectively. Therefore, the preferential priority order of adsorption on (110) plane were C, Cl_2_ and CO. When the Cl_2_ was co-adsorbed with C and CO on the (110) plane, the chemical bond of Cl_2_ extended and was broken, and two Cl atoms were formed and bonded with Zr atoms, with bond lengths of 2.48 Å and 2.44 Å (Figure 6a,b), as well as 2.44 Å and 3.10 Å (Figure 6c,d). After co-adsorbing on the (100) plane with Cl_2_ molecules, the C atoms were transferred to the O1 atom near Cl, the C1-O1 bond was formed with a bond length of 1.19 Å, and the interatomic spacing between C atoms and Cl atoms was 2.44 Å. The CO molecule adsorbed on the (110) plane was connected to the O atom and formed a bond with an interatomic spacing of 1.35 Å, which was longer than that of the C1-O1 bond. The CO molecules migrated towards Cl atoms and formed C-Cl bonds with an interatomic spacing of 1.79 Å, indicating that the interatomic spacing between CO molecules and Cl atoms is shorter and the interaction is stronger. The adsorption energies of the co-adsorption structures formed by C and Cl_2_, and CO and Cl_2_, on the (110) plane were −6.91 eV and −2.31 eV, respectively; therefore, the co-adsorption structure formed by C and Cl_2_ was more stable for promoting the dissociation of Cl_2_. Compared with the (100) plane (−6.61 eV), when C and Cl_2_ were co-adsorbed on the (110) plane, the adsorption energy was lower and the structure was more stable.

Figure 6 shows the differential charge density diagram of the co-adsorption of C and Cl_2_ (Figure 6a,c), as well as CO and Cl_2_ (Figure 6b,d), on the (110) plane (molar ratio C/Cl_2_ = 1). The C-O bonds were observed in the 110-C-Cl_2_ structure, while Cl-C-O bonds were observed in the 110-CO-Cl_2_ structure. Zr-Cl bonds were observed in both co-adsorption structures. An area of charge accumulation was observed around the C-O-Si bond, which implied that C was a donor during the bonding process, and exhibited the donor characteristics. After the dissociation of Cl_2_ molecules, electron-depleted regions were observed around Zr-Cl bonds, and Cl atoms transferred electrons to the (110) plane through the Zr atoms.

Table 2 summarizes the Bader charges of the C, CO and Cl_2_ adsorbed on the (110) plane with a molar ratio of C to Cl_2_ of 1:1. In the co-adsorption structure formed by C and Cl_2_, the electrons released by C to the (110) plane were 1.13 e, and the electrons accepted by Cl atoms from the (110) plane were 0.54 e and 0.62 e, respectively. Compared with the (100) plane, both the electrons released by C atoms and the electrons accepted by Cl atoms increased. For the co-adsorption structure formed by CO and Cl_2_, the electrons accepted by the O atom from the (110) plane were 1.05 e, and the electrons released by the C atom to the (110) plane were 1.76 e. When the Cl atom interacted with the (110) plane, the electrons accepted from the (110) plane were 0.17 e and 0.67 e. This indicated that the CO molecule needed to release more electrons to the (110) plane to form the CO_2_ molecule, and there was a strong interaction between the CO molecule and the (110) plane.

Figure 7 shows the partial density of states of the C and Cl_2_ (Figure 7a) co-adsorption and CO and Cl_2_ (Figure 7b) co-adsorption on the (110) plane (molar ratio C/Cl_2_ = 1). For the co-adsorption structure formed by C and Cl_2_, the resonance peaks related to the Zr-Cl bond were observed on both sides of the Fermi energy level (−6.15 eV, −2.57 eV, −1.42 eV, 2.79 eV, 4.39 eV and 5.67 eV), while the C-O bond’s resonance peak bands were observed at −7.04 eV, −6.15 eV, −1.42 eV and 2.79 eV and 3.91 eV. The resonance peaks related to Si-O bonds were observed on the left (−16.69 eV, −6.15 eV and −4.62 eV) and right side (2.79 eV, 5.48 eV and 4.52 eV) of the Fermi energy level. However, for the co-adsorption structure formed by CO and Cl_2_, the resonance peaks related to Zr-Cl bonds shifted to −7.96 eV, −4.76 eV, 2.91 eV and 5.46 eV. The resonance peaks observed at −7.96 eV, −4.76 eV and 3.55 eV were related to the bonding interactions between C and O atoms. In addition to the left side (−18.18 eV, −4.76 eV and −1.95 eV), resonance peaks were also observed on the right side (3.22 eV, 8.15 eV) of the Fermi energy level, which was related to the bonding interaction between Si and O atoms.

Figure 8 shows the reaction pathways and energy barrier of C and Cl_2_ (a–c), as well as that of CO and Cl_2_ (d–f), co-adsorbed on the (110) plane with a molar ratio of C to Cl_2_ of 1:1. When C and Cl_2_ were adsorbed on the (110) plane, the adsorbed molecules needed to overcome an energy barrier of 0.08 eV to reach the transition state, then the generated CO was desorbed, accompanied by the release of 1.08 eV energy. When CO and Cl_2_ were adsorbed on the (110) plane, the energy barrier between the initial and transition states (forming CO_2_ molecules) was 0.51 eV, and the energy released from the transition state to the final state (CO_2_ molecules desorbed from the surface) was 1.11 eV. The above results indicated that the energy barriers for C and Cl_2_ adsorption and reaction at the (110) plane were lower than CO, thus the carbochlorination reaction was more active. Compared to the (100) plane (0.22 eV and 1.32 eV), the energy barriers for the co-adsorption of C and CO with Cl_2_ on the (110) plane were lower, leading a higher probability of adsorption and reaction occurring on the (110) plane.

The above results indicate that the co-adsorption structure of 110-C-Cl_2_ was more stable, and the energy barriers for C and Cl_2_ adsorption and reaction at (110) were lower, thus the adsorption and reaction have a higher probability of occurring on the (110) plane.

#### 3.4.2. Co-Adsorption on the (110) Plane with a Molar Ratio of C to Cl_2_ of 1:2 and 2:2

According to the co-adsorption and reaction results of C and Cl_2_, as well as CO and Cl_2_, on the (100) and (110) plane, the C was more prone to adsorption and reaction on the surface of ZrSiO_4_ than CO, and the energy barrier of the adsorbent reaction on the (110) surface was lower. Therefore, the next focus will be on the co-adsorption and reaction of C and Cl_2_ on the (110) plane when the molar ratios of C and Cl_2_ were 1:2 (Figure 9a,b) and 2:2 (Figure 9c,d). When the molar ratio of C and Cl_2_ was 1:2, two Cl atoms interacted with Zr atoms to form chemical bonds with bond lengths of 2.48 Å and 2.44 Å, and the interatomic spacing between Cl atoms was 8.80 Å. The C atom transferred from O1 on the (110) plane of ZrSiO_4_ to the O_2_ near Cl, then the C-O_2_ and C-Cl chemical bonds with lengths of 1.19 Å and 2.44 Å were formed. Changing the molar ratio of C and Cl_2_ to 2:2, two C and Cl_2_ were adsorbed on the (110) plane, and the Cl-Cl chemical bond was broken, dissociating into two Cl atoms; the interatomic spacing increased to 4.55 Å, due to the adsorption of one C and Cl_2_ molecule on the original (110) plane, and the separate adsorption of another C and Cl_2_ molecule. The chemical bond of the Cl_2_ molecule was broken, then dissociated into two Cl atoms, and the distance increased to 4.55 Å. Subsequently, the Cl3 and Cl4 were connected to the surface Zr3 and Zr4 atoms to form bonds. The bond lengths of Zr3-Cl3 and Zr4-Cl4 were 2.48 Å and 2.47 Å, respectively. The distance between O_2_ and Zr_2_ atoms increased to 2.60 Å. The C2 atom was connected to the O_2_ atom on the (110) plane and formed a bond, causing the Zr3-O2 bond to break and the spacing to increase to 3.75 Å, while the C2-O2 bond was 1.19 Å, which indicated increased C and Cl_2_, leading to more bonding between the C and Cl atoms. Changing the molar ratio of C and Cl_2_ from 1:1 to 1:2, the co-adsorption energy was increased from −6.91 eV to −6.66 eV, and no significant changes occurred. When the ratio of C and Cl_2_ molecules was increased to 2:2, the co-adsorption energy decreased to −13.45 eV, indicating that increasing the amount of elemental C significantly enhanced the stability of the co-adsorption structure, which was beneficial for promoting the carbochlorination reaction.

Table 3 shows the Bader charge of C and Cl_2_ co-adsorption on the (110) plane with the molar ratios of C to Cl_2_ of 1:2 and 2;2. When the molar ratio of C and Cl_2_ was 1:2, the electrons released by C were 1.14 e, and the electrons accepted from the (110) plane were 0.55 e, 0.60 e, −0.20 e and 0.64 e. Compared with the 1:1 ratio (the molar ratio of C to Cl_2_), the electrons released by C had no significant change, indicating that increasing the Cl_2_ content had no significant effect on the electron transfer and reaction of C. When the molar ratio of C and Cl_2_ was 2:2, the electrons released by C to the (110) plane were 1.17 e and 1.13 e, while the electrons accepted by Cl atoms were 0.53 e, 0.61 e, 0.63 e and 0.60 e. Both the electrons released by the carbon atom and accepted by the Cl atom increased, which indicated that increasing the C content was conducive to promoting electron transfer and carbochlorination reaction.

Figure 10 is the differential charge density diagram for C and Cl_2_ co-adsorption on the (110) plane with ratios of C to Cl_2_ of 1:2 (Figure 10a,b) and (Figure 10c,d). The Zr-Cl and C-O bonds were observed in both structures, and the charge accumulation regions around the C-O-Si bond and the electron-depleted region around the Zr-Cl bond were observed, indicating that when C and Cl_2_ were co-adsorbed on the (110) plane, charge transfer occurred; C output electrons, while Cl atoms transferred electrons to the (110) plane through Zr atoms. This was similar to the results of the 1:1 ratio and the co-adsorption of C and Cl_2_ on the (100) plane.

Figure 11 shows that the partial density of states of adsorption on the (110) plane with ratios of C to Cl_2_ of 1:2 (Figure 11a) and 2:2 (Figure 11b). Resonance peaks related to Zr-Cl, C-O and Si-O bonds were observed in both structures. The resonance peaks related to Zr-Cl bonds were observed at −6.53 eV, −3.66 eV, −3.66 eV, 2.73 eV, 4.07 eV and 5.09 eV, as shown in Figure 11a. In addition, −7.17 eV, −6.59 eV and −1.07 eV resonance peaks related to bonds between C and O atoms were also observed at 2.79 eV and 3.88 eV. Resonance peaks related to bonding between Si and O atoms were observed on the left side (−16.76 eV, −5.89 eV and −4.48 eV) and right side (4.07 eV and 5.61 eV) of the Fermi energy level. When the molar ratio of C to Cl_2_ was 2:2 (Figure 11b), the resonance peaks related to Zr-Cl bonds were also observed at 2.98 eV, 4.53 eV and 5.32 eV, with the exception of −2.03 eV and −0.72 eV. The resonance peaks observed at −8.02 eV, −4.97 eV, −1.36 eV, 2.33 eV and 4.52 eV were related to the bonding between C and O atoms. The resonance peaks related to Si and O atoms were observed on both sides of the Fermi level (−18.29 eV, −6.08 eV, −5.48 eV, 2.81 eV and 7.59 eV).

Figure 12 is the reaction pathway and energy barrier of C and Cl_2_ on the (110) plane with a molar ratio of C and Cl_2_ of 1:2. The CO molecules were formed by C and O, and reached a transition state after overcoming the energy barrier of 0.01 eV. The O1-Si1 bond was broken and produced CO molecules, which then moved away from the (110) surface and released 1.21 eV of energy.

Figure 13 is the reaction pathway and energy barrier of C and Cl_2_ on the (110) plane with a molar ratio of C and Cl_2_ of 2:2. The transition state TS1 was reached after overcoming the energy barrier of 0.02 eV, the O2-Si2 bond was broken to produce the first CO molecule. Subsequently, the CO molecule desorbed from the (110) surface and released 1.42 eV of energy. When C and Cl_2_ were adsorbed on the (110) plane, the formed IM2 was used as the initial structure for removing the second CO molecule. The adsorption energy of this structure was set to 0 eV; after reaching the transition state TS2, the O1-Si1 bond broke while a second independent CO molecule formed, which then released 1.16 eV of energy to reach the final state. The energy barrier between the initial state IM2 and the transition state TS2 was 0.06 eV. indicating that the formation and desorption of the second CO molecule was more difficult.

Based on the results of the reaction energy barrier of C and Cl_2_ on the (110) plane, and compared with the 1:1 ratio (0.08 eV), the co-adsorption energy and reaction energy barrier decreased with the increased amount of C (1:2 and 2:2), resulting in an increase in reaction activity. Increasing the molar ratio of C to Cl_2_ to 2:2, the reaction energy barrier (0.02 eV) for the formation of the first CO molecule did not change significantly, but the co-adsorption energy was significantly reduced and the adsorption stability was enhanced. This indicated that increasing the C content was beneficial for the formation of vacancies, which can adsorb more Cl_2_ molecules and promote the progress of the carbochlorination reaction.

### 3.5. Experimental Verification of Simulation Calculation Results

The above results indicated that increasing the content of C was beneficial for promoting the adsorption of Cl_2_ and charge transfer, reducing the reaction energy barrier, and enhancing the activity of the carbochlorination reaction. In order to ensure the reliability of simulation calculations and to analyze the mechanism of C in the carbochlorination process of ZrSiO_4_, a thermogravimetric analysis was conducted on different powders. After heat treatment, the elemental composition of the surface of ZrSiO_4_ powders was researched.

Figure 14 shows the thermogravimetric curves of a mixture of ZrSiO_4_ and petroleum coke with different molar ratios. The ZrSiO_4_ did not lose much weight in the range of 298–1473 K under nitrogen atmosphere. As the content of petroleum coke increased, the weight loss of the material significantly increased. For example, when the temperature was 1473 K and the ratio of petroleum coke to ZrSiO_4_ was 1:1, the weight loss was 2.15%, and when the ratio was increased to 4:1, the weight loss increased to 3.61%.

The mixed powder of ZrSiO_4_ and petroleum coke was heat-treated under a vacuum atmosphere (1473 K/1h), then the morphology and the surface elemental composition of the heat-treated ZrSiO_4_ powder were observed by SEM. Figure 15a shows the surface morphology of ZrSiO_4_ with a molar ratio of ZrSiO_4_ to petroleum of 1:1. The element composition on the surface of ZrSiO_4_ (site A in Figure 15a) was calculated by quantitative analysis from the EDS spectrum, and, as shown in Figure 15c, the atomic percentage of surface O was 65.84%, which was lower than that of ZrSiO_4_ (66.66%). The morphology of heat-treated ZrSiO_4_ powder with a molar ratio of ZrSiO_4_ to petroleum of 1:4 is shown in Figure 15b, and Figure 15d–f shows the surface (B, C and D) element composition. The surface oxygen content of heat-treated ZrSiO_4_ powder decreased with the increased C content. When the molar ratio of ZrSiO_4_ to petroleum coke was 1:4, the content of oxygen atoms on the B, C and D regions of ZrSiO_4_ decreased to 59.35%, 47.02% and 51.62%, respectively, while the content of Si and Zr increased.

Based on the results of adsorption energy, Bader charge, reaction energy barrier, thermogravimetric analysis and EDS, increasing the carbon content is beneficial for the adsorption of C and Cl_2_ on the surface of ZrSiO_4_, reducing the reaction energy barrier, promoting electron transfer, reducing oxygen atoms on the surface of ZrSiO_4_ and increasing unbound Zr atoms and vacancies, which can promote the breaking of Cl-Cl bonds, the formation of Zr-Cl bonds and the carbochlorination reaction of ZrSiO_4_.

## 4. Conclusions

Based on first principles, the adsorption structure, adsorption energy, charge density, density of states and reaction path properties of the co-adsorption system were analyzed. The co-adsorption behavior of C, CO and Cl_2_ on ZrSiO_4_ (100) and (110) surfaces was studied, and the conclusions are as follows:(1)When C, CO and Cl_2_ co-adsorbed on the (100) and (110) surfaces, they interacted with the surface, the Cl_2_ molecules dissociated, and then the Zr-Cl and C=O bonds formed on the surface of ZrSiO_4_.(2)Compared with CO, the co-adsorption energy of C and Cl_2_ on the (100) and (110) planes of ZrSiO_4_ was lower, and the co-adsorption structure was more stable. Adding Cl_2_ molecules did not result in a significant change in co-adsorption energy, while increasing the C content resulted in a significant decrease in co-adsorption energy. The co-adsorption energy of the 110-2C-2Cl_2_ structure was the lowest (−13.45 eV), and the co-adsorption structure was the most stable.(3)When C, CO and Cl_2_ co-adsorbed on the (100) and (110) planes, C provided electrons, while O and Cl accepted electrons. The charge aggregation regions were formed around Si-O and C-O-Si bonds, while charge-depleted regions were formed around the Zr-Cl bond. C can promote the co-adsorption, charge transfer and reaction between Cl_2_ and the surface; increasing the C content was beneficial for promoting co-adsorption and reaction with Cl_2_ on the surface of ZrSiO_4_, removing surface oxygen atoms, which is consistent with the experimental results.(4)Compared with CO, the reaction energy barrier of C and Cl_2_ was lower. The higher the C content, the lower the adsorption energy barrier, which was beneficial for promoting the charge transfer and dissociation of Cl_2_.

## Figures and Tables

**Figure 1 materials-17-01500-f001:**
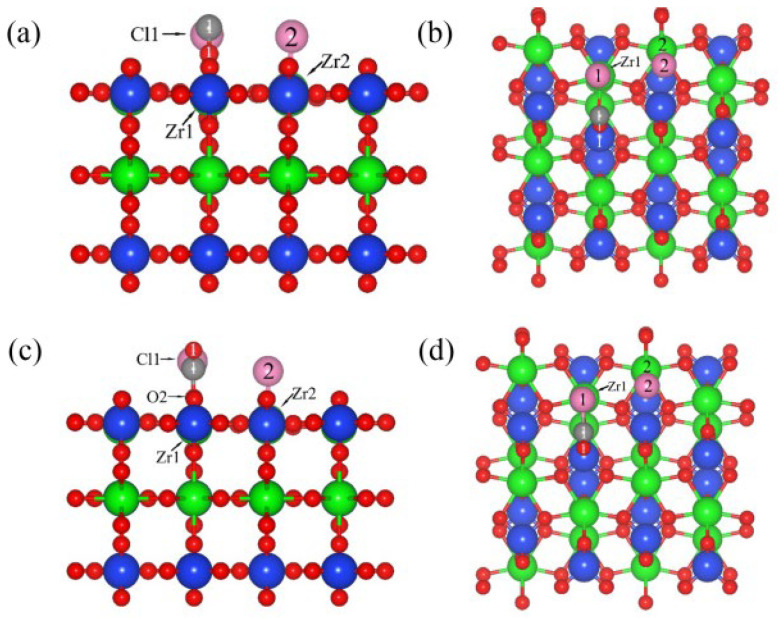
Optimized model structure for the C and Cl_2_ (**a**,**b**) co-adsorption and CO and Cl_2_ (**c**,**d**) co-adsorption on the (100) plane: front view (**a**,**c**) and top view (**b**,**d**).

**Figure 2 materials-17-01500-f002:**
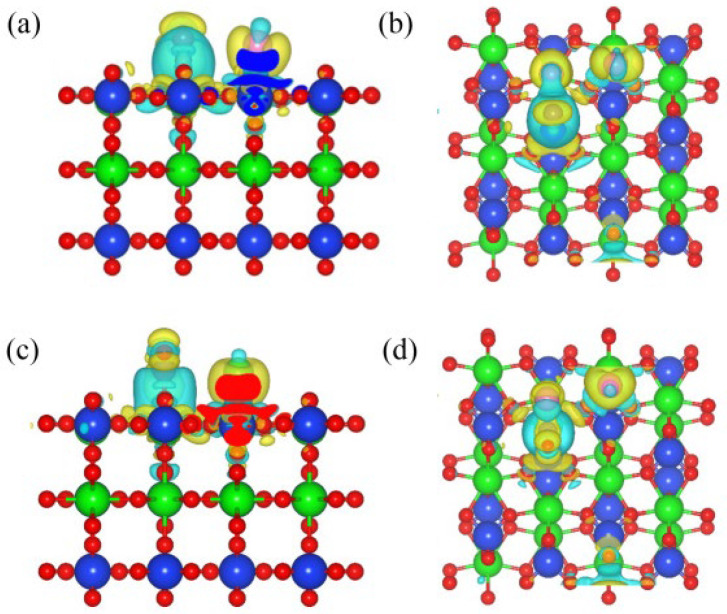
Differential charge density diagram of the C and Cl_2_ (**a**,**b**) co-adsorption and CO and Cl_2_ (**c**,**d**) co-adsorption on the (100) plane: front view (**a**,**c**) and top view (**b**,**d**).

**Figure 3 materials-17-01500-f003:**
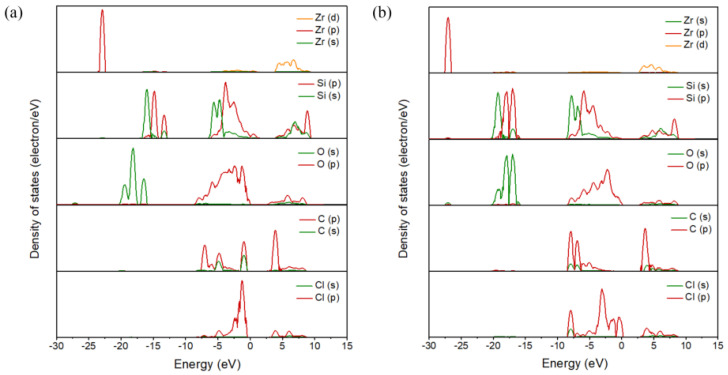
Projected density of state of the C and Cl_2_ (**a**) co-adsorption and CO and Cl_2_ (**b**) co-adsorption on the (100) plane.

**Figure 4 materials-17-01500-f004:**
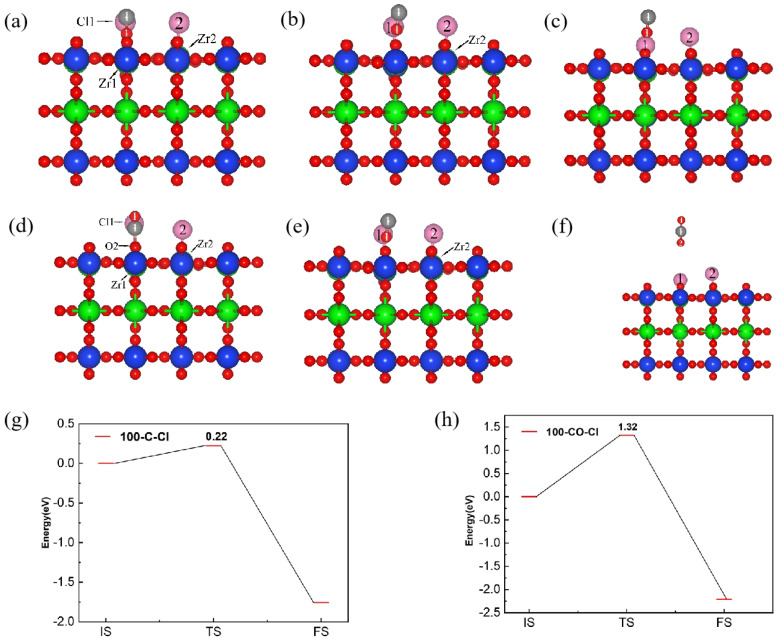
The front view of initial state (**a**,**d**), transition state (**b**,**e**) and final state (**c**,**f**) of the reaction pathway and energy barrier of (**g**,**h**) C and Cl_2_ (**a**–**c**,**g**) as well as CO and Cl_2_ (**d**–**f**,**h**), on the (100) plane.

**Figure 5 materials-17-01500-f005:**
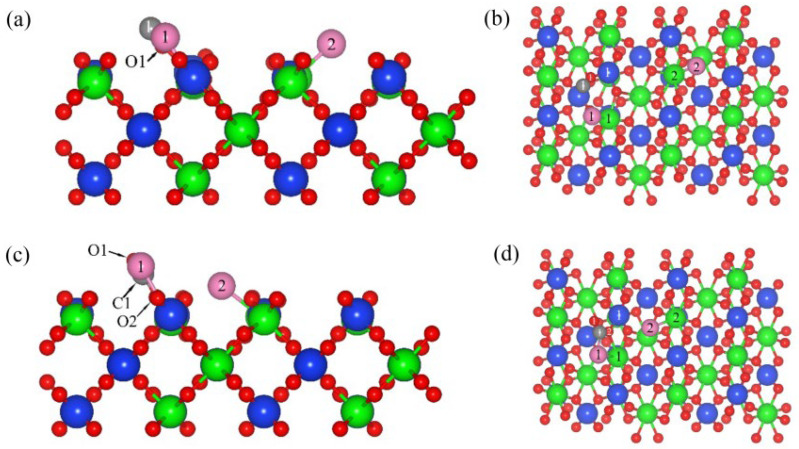
Optimized model structure for the C and Cl_2_ (**a**,**b**) co-adsorption and CO and Cl_2_ (**c**,**d**) co-adsorption on the (110) plane with a molar ratio of C to Cl_2_ of 1:1: front view (**a**,**c**) and top view (**b**,**d**).

**Figure 6 materials-17-01500-f006:**
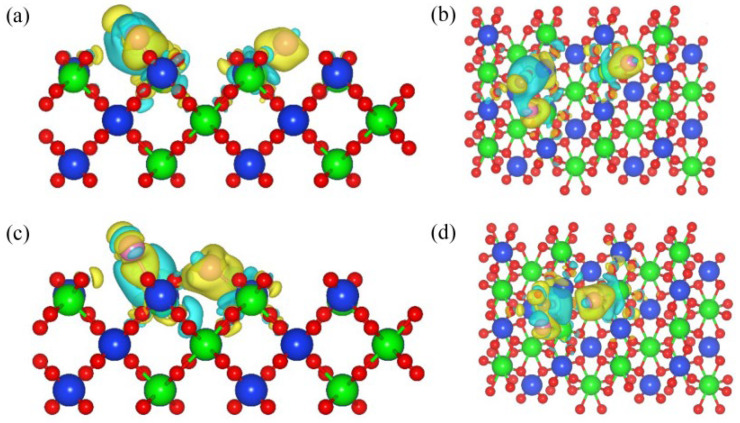
Differential charge density diagram of the C and Cl_2_ (**a**,**b**) co-adsorption and CO and Cl_2_ (**c**,**d**) co-adsorption on the (110) plane with a molar ratio of C to Cl_2_ of 1:1: front view (**a**,**c**) and top view (**b**,**d**).

**Figure 7 materials-17-01500-f007:**
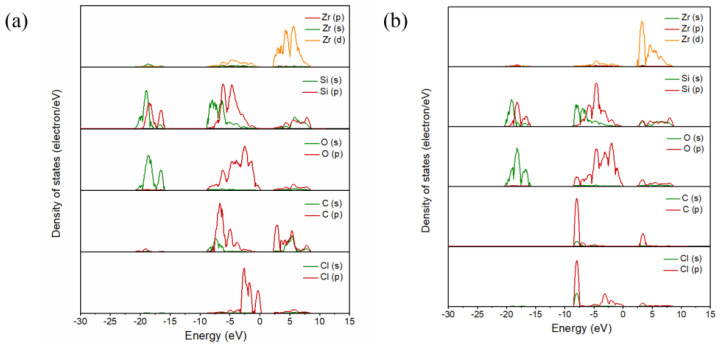
Projected density of state of the C and Cl_2_ (**a**) co-adsorption and CO and Cl_2_ (**b**) co-adsorption on the (110) plane with a molar ratio of C to Cl_2_ of 1:1.

**Figure 8 materials-17-01500-f008:**
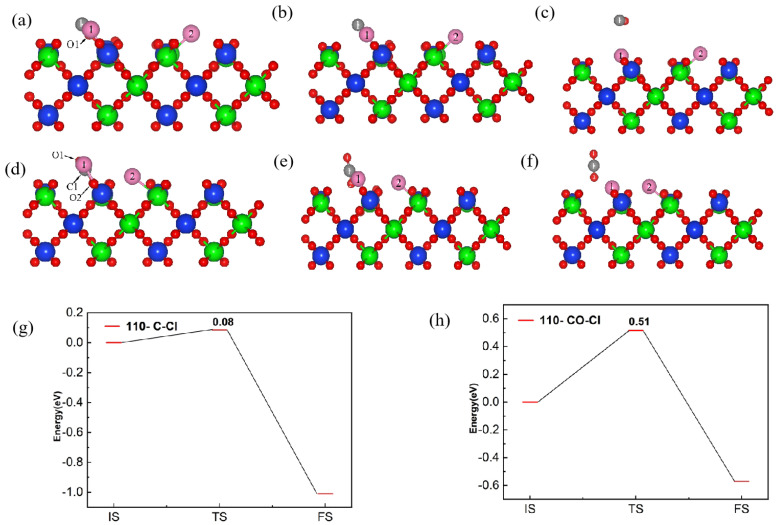
The front view of initial state (**a**,**d**), transition state (**b**,**e**) and final state (**c**,**f**) of the reaction pathway and energy barrier of (**g**,**h**) C and Cl_2_ (**a**–**c**,**g**), as well as CO and Cl_2_ (**d**–**f**,**h**), on the (100) plane and on the (110) plane with a molar ratio of C to Cl_2_ of 1:1.

**Figure 9 materials-17-01500-f009:**
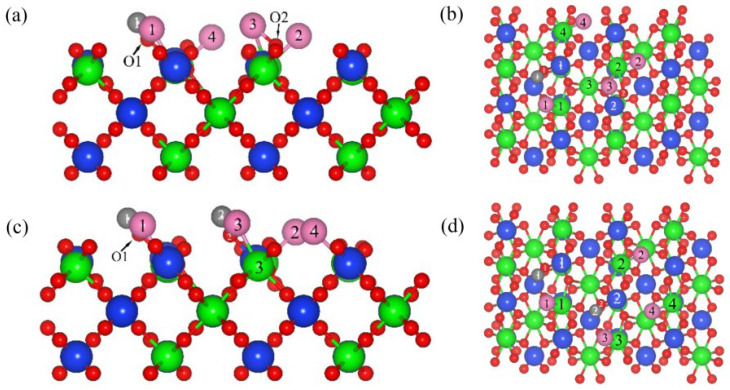
Optimized model structure for C and Cl_2_ co-adsorption on the (110) plane with a ratio of C to Cl_2_ of 1:2 (**a**,**b**) and 2:2 (**c**,**d**): front view (**a**,**c**) and top view (**b**,**d**).

**Figure 10 materials-17-01500-f010:**
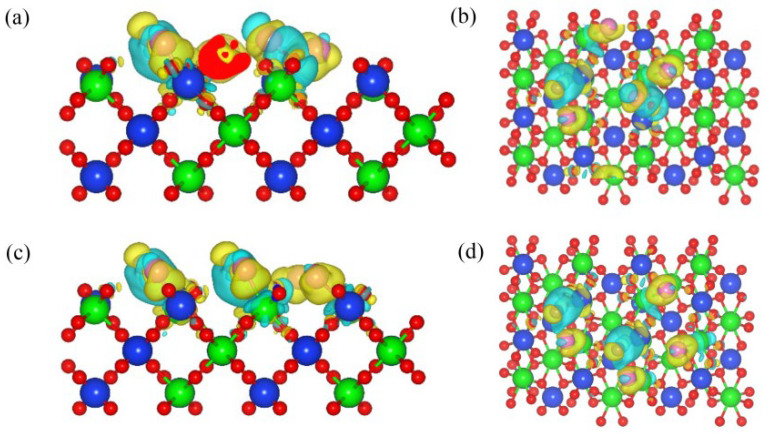
Differential charge density diagram for C and Cl_2_ co-adsorption on the (110) plane with ratios of C to Cl_2_ of 1:2 (**a**,**b**) and (**c**,**d**): front view (**a**,**c**) and top view (**b**,**d**).

**Figure 11 materials-17-01500-f011:**
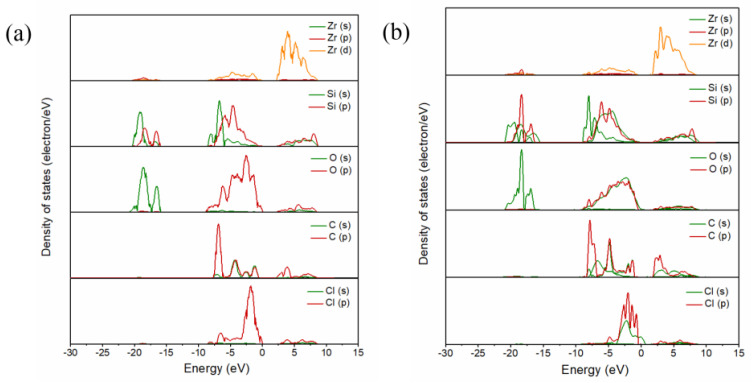
Projected density of state of the C and Cl_2_ co-adsorption on the (110) plane with molar ratios of C to Cl_2_ of 1:2 (**a**) and 2:2 (**b**).

**Figure 12 materials-17-01500-f012:**
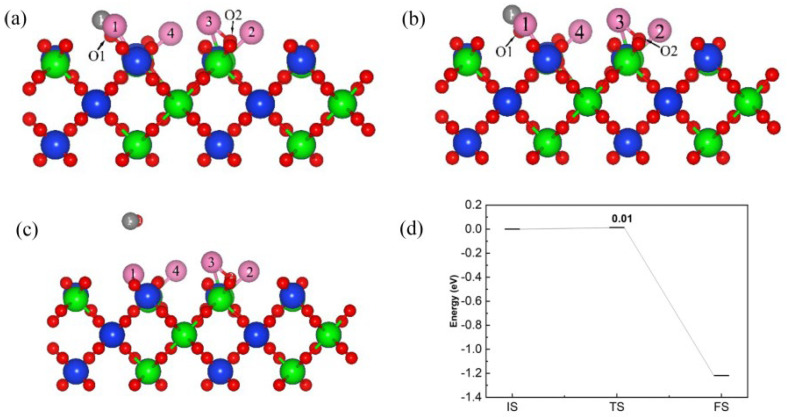
The front view of the initial state (**a**), transition state (**b**) and final state (**c**) of the reaction pathway (**d**) for C and Cl_2_ on the (110) plane with a ratio of C to Cl_2_ of 1:2.

**Figure 13 materials-17-01500-f013:**
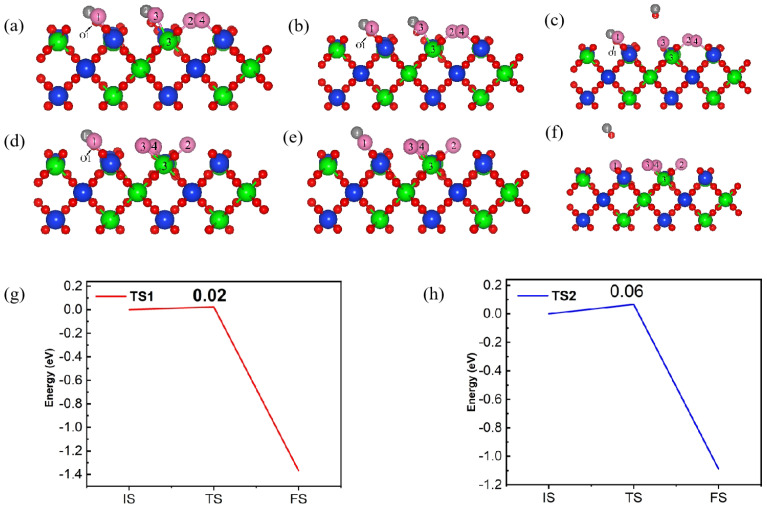
The front view of the initial state (IS1, (**a**), IM2, (**d**)), transition state (TS1, (**b**), TS2, (**e**)), intermediate state (IM1, (**c**)) and final state (**f**) of reaction pathway (**g**,**h**) for C and Cl_2_ on (110) plane with the ratio of C to Cl_2_ was 2:2.

**Figure 14 materials-17-01500-f014:**
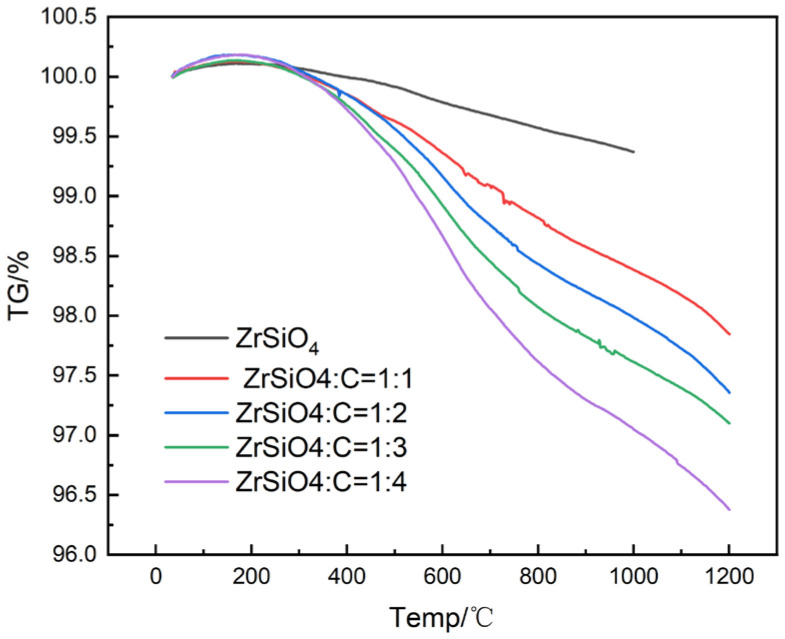
The thermogravimetric analysis results of materials with different proportions.

**Figure 15 materials-17-01500-f015:**
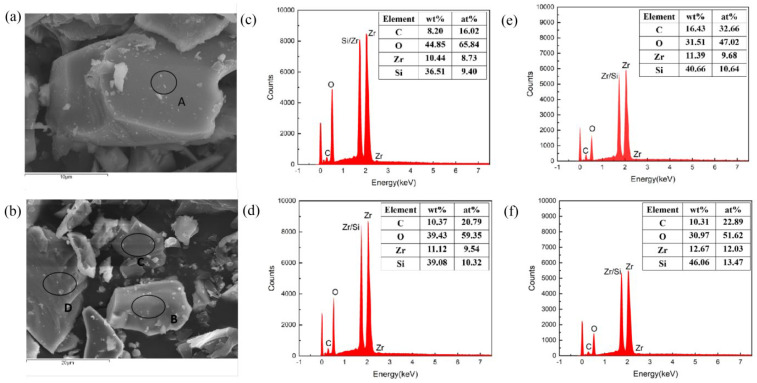
Powder morphology (**a**,**b**) and its surface element composition (**c**–**f**) with ratios of C to ZrSiO_4_ of 1:1 (**a**,**c**) and 4:1 (**b**,**d**–**f**). The A is the surface of heat-treated ZrSiO_4_ powder with a molar ratio of ZrSiO_4_ to petroleum of 1:1, and the B,C,D are the surface of different heat-treated ZrSiO_4_ powder with a molar ratio of ZrSiO_4_ to petroleum of 1:4.

**Table 1 materials-17-01500-t001:** Bader charge of C, CO and Cl_2_ co-adsorption on the (100) plane.

Adsorption Structure	Element	Bader Charge
100-C-Cl_2_	C	−1.06
	Cl	0.44
	Cl	0.62
100-CO-Cl_2_	O	1.05
	C	−1.78
	Cl	0.16
	Cl	0.64

**Table 2 materials-17-01500-t002:** Bader charge of C, CO and Cl_2_ co-adsorption on the (110) plane with a molar ratio of C to Cl_2_ of 1:1.

Adsorption Structure	Element	Bader Charge
110-C-Cl_2_	C	−1.13
	Cl	0.54
	Cl	0.62
110-CO-Cl_2_	O	1.05
	C	−1.76
	Cl	0.17
	Cl	0.67

**Table 3 materials-17-01500-t003:** Bader charge of C and Cl_2_ co-adsorption on the (110) plane of ZrSiO_4_ with molar ratios of C to Cl_2_ of1:2 and 2:2.

Adsorption Structure	Element	Bader Charge
110-C-2Cl_2_	C	−1.14
	Cl	0.55
	Cl	0.60
	Cl	−0.20
	Cl	0.64
110-2C-2Cl_2_	C	−1.13
	C	−1.17
	Cl	0.53
	Cl	0.61
	Cl	0.63
	Cl	0.60

## Data Availability

Data are contained within the article and Appendix A.

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
