# Peer review of "First Principles Investigation of C, Cl2 and CO Co-Adsorption on ZrSiO4 Surfaces for Carbochlorination Reaction"

_materials, 2024, doi:10.3390/ma17071500_

Round 1
Reviewer 1 Report
Comments and Suggestions for Authors
The paper, "First Principles Investigation of C, Cl2, and CO Co-adsorption on ZrSiO4 Surfaces for Carbochlorination Reaction," presents a detailed study of the adsorption behaviors and mechanisms of carbon, chlorine, and carbon monoxide on zirconium silicate surfaces. It leverages first-principles calculations to analyze adsorption structures, energies, charge transfers, and reaction mechanisms, aiming to enhance the understanding of carbochlorination reactions crucial for zirconium oxychloride production. The study concludes that co-adsorption of C and Cl2 on ZrSiO4 surfaces is favorable for the reaction, with increased carbon content improving adsorption and reaction efficiencies.
However, the paper need be revised before publication.
- The paper does not explicitly address potential experimental validation of theoretical findings. The paper contains an experimental part, but it is practically unrelated to the conducted calculations and looks more like a foreign piece. It is necessary to bridge the experimental and theoretical parts.
- The comparison of the obtained results with existing literature is clearly insufficient.
- The novelty of the work is not clearly stated.
- the name of the program (VASP not VSP simulation software) should be corrected.
- When describing k points used to sample the Brillouin zone (5×5×1 and 4×3×1, etc.), it is advisable to use standard terminology https://www.vasp.at/wiki/index.php/KPOINTS.
- "cutoff energy on the ZrSiO4 (110) surface were set to 3×2×1 and 480 eV" - in this phrase, only 480 eV refers to the cutoff energy. 3×2×1 is about the k-point sampling.
- The model for ZrSiO4 Surfaces (slab thickness, vacuum gap) is not clear. It must be explicitly described.
- The paper presents calculations of absorption on the surface, describing the situation in the absence of temperature (or 0 K, as sometimes said about quantum chemical calculations). How accurate is the assessment of barriers and other conclusions without subsequent thermodynamic analysis?
Overall, I cannot recommend the paper for publication in its current version, the authors have not properly correlated the results of the proposed model with real life.
Comments on the Quality of English Language
It is necessary to correct both the style and incompleteness of many descriptions (in particular, the section on computational details is very difficult to read due to incomplete descriptions and missing terms).
Author Response
Dear Reviewer:
Thank you for your concerning our manuscript entitled “First Principles Investigation of C、Cl2 and CO Co-adsorption on ZrSiO4 Surfaces for Carbochlorination Reaction”. On behalf of co-authors, We appreciate for their positive and constructive comments and suggestions on our manuscript (Materials-2873091). We have studied your comments carefully and have made major revision which marked with red font in the manuscript. We have tried our best to revise our manuscript according to the comments.
According to your review comments, we first carefully revised our paper, please see the following Response 1 to 8.
Thank you again for your valuable comments and suggestions for revising our articles. we also thank you very much for giving us an opportunity to revise our manuscript again. We hope that our manuscript could be considered for publication in Materials.
Best wishes!

Reviewer 2 Report
Comments and Suggestions for Authors
The authors investigate on the adsorption mechanisms of C, Cl2 and CO on ZrSiO4 surfaces by means of first principles calculations, and attempt to validate experimentally their model. The subject might be of interest to those seeking to understand the reaction mechanisms of ZrSiO4 carbochlorination. The manuscript is however lacking structural clarity, and the methodology has some flows, therefore, before recommending publishing, several issues should be addressed.
1. The introduction fails to give a background regarding the knowledge gaps in ZrSiO4 carbochlorination reaction mechanisms and how this study contributes to filling them (or at least some of them).
2. Although the study envisages specific crystal planes of the ZrSiO4 structure, nowhere in the text references about the type on the structure or the space group can be found. This is a serious methodological issue and should be revised.
3. Instead of a reference to “the new morphology of ZrSiO4 revealed by Korkin et al.” a brief description of the crystal structure is preferable.
4. In the results and discussion section each subsection should have clearly formulated conclusions.
5. The previous point might help to reformulate the conclusions paragraph, which in its current form is partly ambiguous. Also, this paragraph should also resume the experimental findings and how are they fit to the theoretical model.
6. The manuscript should be also carefully checked of grammar errors, typos, unneeded repetitions, etc. Several are given below.
- ZrCl4 [12], in the first paragraph of the introduction;
- “Therefore, which has been”, same paragraph;
- “DUNE et al.[15]”, caps, introduction;
- “The results indicated that there a bonding interactions”, there is, page 5;
- “Based on the results of the co-adsorption energy and reaction energy barrier of C and Cl2 on (110) plane, compared with 1:1 (0.08 eV), the reaction energy barrier decreased with the increased the increasing the amount of C or Cl2 (1:2 and 2:2), resulted in an increase in reaction activity.”, should be reformulated as it is confusing.
- “on first principles principles”, in the conclusions.
Comments on the Quality of English Language
The manuscript should be checked for grammar errors. Several paragraphs should be reformulated.
Author Response
Dear Reviewer:
Thank you for your concerning our manuscript entitled “First Principles Investigation of C、Cl2 and CO Co-adsorption on ZrSiO4 Surfaces for Carbochlorination Reaction”. On behalf of co-authors, We appreciate for their positive and constructive comments and suggestions on our manuscript (Materials-2873091). We have studied your comments carefully and have made major revision which marked with red font in the manuscript. We have tried our best to revise our manuscript according to the comments.
Thank you again for your valuable comments and suggestions for revising our articles. we also thank you very much for giving us an opportunity to revise our manuscript again. We hope that our manuscript could be considered for publication in Materials.
Best wishes!

Reviewer 3 Report
Comments and Suggestions for Authors
The manuscript "First Principles Investigation of C、Cl2 and CO Co-adsorption on ZrSiO4 Surfaces for Carbochlorination Reaction" studies the absorption of C, Cl2 and CO by ZrSiO4 (100) and (110) surfaces. The authors calculated the Barder charge that occurs due to absorption, they analyzed the adsorption structure, adsorption energy, charge density, density of states, and reaction path properties of the co-adsorption system. They observed based on the results of adsorption energy, Bader charge, reaction energy barrier, thermogravimetric analysis, and EDS, that increasing the carbon content is beneficial for the adsorption of C and Cl2 on the surface of ZrSiO4. The experimental studies were provided on ZiSiO4 powder mixed with petroleum coke with a 1:1 and 1:4 ratio of C to ZiSiO4. However, it is not clear if the experiment supports this conclusion.
Author Response

(The authors gave the same response as above.)

Reviewer 4 Report
Comments and Suggestions for Authors
The authors performed a theoretical investigation using DFT on C, Cl2, and CO co-adsorption on ZrSiO4 Surfaces for Carbochlorination Reaction. The authors focused on the (100) and (110) surfaces as research objects, and investigated the co-adsorption behavior of C, CO, and Cl2 on the (100) and (110) surfaces of ZrSiO4 at the atomic level, based on the first principles calculations, which can assist in understanding the reaction mechanism of ZrSiO4 carbochlorination. The authors describe an experimental part in their manuscript.
I believe this work requires some necessary changes before considering its publication.
-The authors must improve the figures. The insets are too small to follow; the letterings are difficult to read.
-When the authors mention “front view”, do they mean side view?
- When the authors mention “top view”, do they mean from above?
-The authors must show clearly the initial configuration of the system before the adsorption process. What are the initial distances from the molecules to the surface?
-The authors must mention the aim of the experimental part.
-The agreement or disagreement of their experiment should be mentioned in the conclusions.
Comments on the Quality of English Language
The English language requires moderate editing.
Author Response

(The authors gave the same response as above.)

Round 2
Reviewer 4 Report
Comments and Suggestions for Authors
The authors have made almost all the requested changes, except one, the authors must show clearly the initial configuration of the system before the adsorption process. What are the initial distances from the molecules to the surface? They did not show the initial distances of the molecules C, CO, and Cl2 from the surface before the adsorption process.
Comments on the Quality of English Language
Only minor editing of English language is required.
Author Response

(The authors gave the same response as above.)
